# A Transformer-Based Channel Estimation Method for OTFS Systems

**DOI:** 10.3390/e25101423

**Published:** 2023-10-07

**Authors:** Teng Sun, Jiebiao Lv, Tao Zhou

**Affiliations:** 1The 54th Research Institute of CETC, Shijiazhuang 050081, China; 15176948810@126.com; 2School of Electronic and Information Engineering, Beijing Jiaotong University, Beijing 100044, China; 21120105@bjtu.edu.cn

**Keywords:** OTFS, channel estimation, deep learning, transformer

## Abstract

Orthogonal time frequency space (OTFS) is a novel modulation scheme that enables reliable communication in high-mobility environments. In this paper, we propose a Transformer-based channel estimation method for OTFS systems. Initially, the threshold method is utilized to obtain preliminary channel estimation results. To further enhance the channel estimation, we leverage the inherent temporal correlation between channels, and a new method of channel response prediction is performed. To enhance the accuracy of the preliminary results, we utilize a specialized Transformer neural network designed for processing time series data for refinement. The simulation results demonstrate that our proposed scheme outperforms the threshold method and other deep learning (DL) methods in terms of normalized mean squared error and bit error rate. Additionally, the temporal complexity and spatial complexity of different DL models are compared. The results indicate that our proposed algorithm achieves superior accuracy while maintaining an acceptable computational complexity.

## 1. Introduction

Currently, orthogonal frequency division multiplexing (OFDM) is the main communication modulation technique used in long-term evolution (LTE) and fifth-generation (5G) systems because of its excellent spectrum utilization [1]. However, OFDM is highly susceptible to the effects of Doppler spread, as it relies on maintaining orthogonality between subcarriers [2]. Consequently, inter-subcarrier interference becomes prevalent in high-mobility communication scenarios within an OFDM system [3]. To address this issue, a novel modulation method called orthogonal time frequency space (OTFS) was introduced in [4], specifically designed for high-mobility environments. The OTFS method extends the signal into the time–frequency (TF) domain and introduces the delay-Doppler (DD) domain through a two-dimensional transformation, thus modulating the signal in the DD domain [5]. This transformation offers the advantage of converting the double-dispersive channel in the TF domain into a time-invariant channel in the DD domain. As a result, all the symbols within an OTFS frame encounter nearly identical sparse channels, reducing the channel estimation overhead in time-varying channels and eliminating the need for high-density pilots to estimate the channel response [6].

Accurately estimating the channel in the delay-Doppler (DD) domain is crucial for successful signal detection at the receiver. In [7], the authors presented a channel estimation algorithm that relies on a single pilot symbol. To enhance the accuracy of the channel estimation, the authors in [7] incorporated a protection interval around the pilot to mitigate interference between the pilot symbol and data symbols. Within this protection interval, channel estimation was performed. Additionally, a data-aided channel estimation scheme was proposed in [8]. In this particular scheme, the pilot symbols are directly embedded within the data symbols without the inclusion of a separate protection interval. This approach leads to improved spectrum utilization while maintaining a similar bit error rate (BER) performance compared to the scheme described in [7]. A new pilot pattern was introduced in [9], building upon the single pilot scheme proposed in [7]. The authors optimized the pilot pattern by removing the protection symbols on the right side of the pilot symbol. Since the data symbols on the right side do not influence the pilot in terms of delay domains, eliminating the protection symbols in this region enhances spectral efficiency effectively. Apart from the single pilot channel estimation techniques, another approach was introduced by the authors in [10], which involved a channel estimation method based on compressed channel sensing. This method takes advantage of the sparsity property of the channel in the DD domain and converts the channel estimation task into signal recovery. By doing so, this method achieves more precise channel estimation while reducing the required pilot overhead. A high-performance algorithm with low complexity is presented in [11]. The memory approximate message passing (AMP) detector can exploit the sparsity of the channel matrix and perform only matrix-vector multiplication in each iteration. To alleviate the performance degradation caused by positive reinforcement during the iteration process, the memory AMP detector utilizes all previous information to ensure the orthogonality principle. In [12], the authors propose a scheme for joint channel estimation and data detection in a hybrid reconfigurable intelligent surface-aided millimeter wave OTFS system. The simulation results demonstrate that the proposed method can accurately obtain the channel and the unknown data symbols.

Deep learning (DL) has experienced significant advancements in image and speech recognition, and its application has extended to the field of communication in recent years [13]. In [14], the authors proposed a method that utilizes a deep neural network (DNN) for both channel estimation and signal detection. They trained a neural network model using the received signals and the original transmission data, allowing for online recovery of the transmitted data through the trained model. In [15], the time–frequency response of the fast-fading channel was treated as a two-dimensional image. By leveraging the known values of the pilot, the unknown values of the channel response were reduced, addressing the challenge of estimating the channel in the presence of fast fading. In [16], a multi-layer neural network model based on a recurrent neural network (RNN) was proposed to estimate the DD domain channel with embedded pilots. The simulation results show that this method had better performance than that in [7]. In [17], the authors utilized a deep convolutional neural network (CNN) to address the issue of interference and noise in the channel matrix within the delay-Doppler (DD) domain. By applying the CNN for denoising, they achieved improved channel estimation results compared to existing methods. Additionally, this scheme demonstrated a notable enhancement in spectral efficiency. In [18], a two-dimensional CNN was proposed to accomplish signal detection in OTFS systems. The 2D-CNN can incorporate the imaginary part of the signal during training and the online phase to acquire a three-dimensional array channel. This method has a significant performance improvement over current methods in high-Doppler channels. In [19], deep RNN was used for channel prediction for the MIMO system, the proposed channel predictor contained a long short-term neural network (LSTM) and gated recurrent unit (GRU) to predict the channel state information (CSI). The authors in [20] proposed a 2D-convolutional long short-term memory network (2D-ConvLSTM) to estimate the channel coefficients. In this context, the channel is considered a two-dimensional convolution in the DD domain, which allows for the utilization of the 2D-ConvLSTM network to predict these coefficients. In [21], a deep residual learning network (ResNet) was used to enhance the performance of traditional channel estimation algorithms. The LSTM and implied temporal correlation between channels were used in the channel prediction in [22,23].

In this work, we employ a predictive approach to enhance the performance of channel estimation by considering the implied time correlation. By capturing and utilizing this time correlation information, we aim to make predictions about the channel state and improve the accuracy of the estimation process. First, we obtain preliminary channel estimation results by using the algorithm in the literature [6]. The preliminary results are considered as time series due to the temporal correlation between channels, and then the series are put into the neural network to obtain more accurate channel estimation results. The remainder of this paper is outlined as follows: Section 2 describes the system model and threshold channel estimation algorithm. Then, the proposed channel estimation algorithm based on a transformer neural network is presented in Section 3. Simulation results are provided in Section 4, and the paper concludes in Section 5.

## 2. System Model

### 2.1. Basic Concept

The OTFS modulation block diagram is shown in Figure 1. At the transmitter, data symbols, $x\left[k,l\right]$, of size N×M are placed in the DD domain, and the data symbols are mapped to the TF domain symbols, X[n,m], by the inverse symplectic finite Fourier transform (ISFFT):(1)$$X[n,m]=\frac{1}{\sqrt{MN}}\sum_{k=0}^{N-1}\sum_{l=0}^{M-1}X[k,l]e^{-j2\pi (\frac{ml}{M}-\frac{nk}{N})}$$

After that, X[n,m] is transformed from the TF domain to the time domain by the Heisenberg transform:(2)$$s(t)=\sum_{n=0}^{N-1}\sum_{m=0}^{M-1}X[n,m]e^{j2\pi m\Delta f(t-nT)}p_{tx}(t-nT)$$
where $p_{tx}(t)$ is the pulse shape at the transmitter, and Δf and T are the subcarrier interval and the sampling interval, respectively.

From [3], the channel model in the DD domain can be expressed as
(3)$$h(\tau ,\nu )=\sum_{i=1}^{P}h_{i}\delta (\tau -\tau_{i})\delta (\nu -\nu_{i})$$
where P is the number of paths, and $h_{i},\tau_{i},\nu_{i}$ stand for channel path gain, delay, and Doppler shift, respectively. The time domain signal, r(t), is obtained at the receiver after passing through the channel, which can be written as
(4)$$r(t)=\int_{\nu }\int_{\tau }h(\tau ,\nu )e^{j2\pi v(t-\tau )}s(t-\tau )d\tau d\nu +n(t)$$
where n(t) denotes Gaussian noise.

The Wigner transform converts the received signal, r(t), into the TF domain signal, Y[n,m]:(5)$$A_{p_{rx,r}}(t,f)≜\int p_{rx}^{*}(t^{\prime }-t)r(t)e^{-j2\pi f(t^{\prime }-t)}dt^{\prime }$$
(6)$$Y[n,m]=A_{p_{rx,r}}(t,f)\left|{}_{t=nT,\;f=m\Delta f}\right.$$
where $p_{rx}(t)$ represents the receive pulse shape, and ${\left(•\right)}^{*}$ represents the complex conjugate operation. Finally, the TF domain signal is converted to the DD domain signal by SFFT:(7)$$y[k,l]=\frac{1}{\sqrt{MN}}\sum_{k=0}^{N-1}\sum_{l=0}^{M-1}Y[n,m]e^{-j2\pi (\frac{nk}{N}-\frac{ml}{M})}$$

From (1)–(7), we can obtain the input–output relationship formula in the DD domain:(8)$$y[k,l]=\sum_{k^{\prime }=0}^{N-1}\sum_{l^{\prime }=0}^{M-1}x[k^{\prime },l^{\prime }]h_{\omega }[{\left(k-k^{\prime }\right)}_{N},{\left(l-l^{\prime }\right)}_{M}]+n[k,l]$$
where n[k,l] is the Gaussian noise term with power spectral density, $N_{ο}$, and $h_{\omega }[k,l]$ denotes the equivalent channel matrix in the DD domain. It is worth mentioning that the pulse used in Equation (4) is the ideal pulse. However, in real systems, non-ideal pulses can lead to cases of fractional Doppler [24]. In this paper, we focus on the ideal pulse case, and Equation (8) is only applicable to that specific scenario.

### 2.2. The Threshold Scheme

The pilot placement of the threshold scheme in [7] is shown in Figure 2. For a given maximum delay index, $l_{\max}$, and Doppler index, $k_{\max}$, since each symbol in the DD domain is considered to experience similar fading, only one pilot symbol is needed to complete the channel estimation. The pilot symbol at the transmitter in the DD domain can be represented as
(9)$$x[k,l]=\left\{\begin{array}{cc} x_{p} & k=k_{p,}l=l_{p}\\ 0 & \begin{array}{l} k\in [k_{p}-2k_{\max},k_{p}+2k_{\max}],\\ l\in [l_{p}-l_{\max},l_{p}+l_{\max}] \end{array}\\ x_{d} & otherwise. \end{array}\right.$$
where $x_{p}$ denotes the pilot symbol, $x_{d}$ denotes the data symbols, and we have $N_{n}=(2l_{\max}+1)(4k_{\max}+1)-1$ as guard symbols. 

We can rewrite the input–output relationship of the pilot according to (8) as
(10)$$y\left[k,l\right]=b\left[k-k_{\max},l-l_{\max}\right]\hat{h}\left[k-k_{\max},l-l_{\max}\right]x_{p}+n\left[k,l\right]$$

At the receiver, for the channel estimation part, $k\in [k_{p}-k_{\max},k_{p}+k_{\max}],l\in [l_{p},l_{p}+l_{\max}]$, we set a threshold, $\Gamma =3\sqrt{N_{ο}}$, where $N_{ο}$ represents the power spectral density of noise. If $\left|y[k,l]\right|\geq \Gamma$, we can assume that $b[k-k_{\max},l-l_{\max}]=1$ and $\hat{h}\left[k-k_{\max},l-l_{\max}\right]=y\left[k,l\right]/x_{p}$. Otherwise, the result can be written as $\hat{h}\left[k-k_{\max},l-l_{\max}\right]=n\left[k,l\right]$. This means that if the path exists, the received signal after this algorithm is a pilot containing Gaussian noise. Otherwise, it is only noise.

### 2.3. Minimum Mean Square Error Detection Algorithm

The BER is also a measure of channel estimation performance. We assume that in the received signal, $\hat{x}=Wy$, where W denotes the weighted matrix, the error of estimation is $e=\hat{x}-x=Wy-x$, and the mean square error can be written as $e_{MSE}=E{\left\|Wy-x\right\|}^{2}$. The minimum mean square error (MMSE) detection algorithm minimizes the mean square error between the actual signal and the estimated signal using the minimum mean square error as a criterion. When $e_{MSE}$ is minimized, the weighed matrix can be written as
(11)$$W_{MMSE}=H^{H}{\left(H^{H}H+\sigma^{2}I\right)}^{-1}$$
where H represents the effective channel matrix of size MN×MN, and $\sigma^{2}$ represents the variance of noise. The recovered signal after using this algorithm can be represented as $\hat{x}=W_{MMSE}y$.

## 3. Proposed Transformer Estimation Algorithm

### 3.1. Architecture of the Proposed Channel Estimation Algorithm

Our proposed Transformer estimation structure is shown in Figure 3. The Transformer estimation algorithm is divided into two phases: offline training and online prediction. During the offline training phase, the neural network is trained using a substantial amount of channel data. This training process aims to optimize the parameters of the Transformer neural network for accurate channel estimation. In the subsequent online prediction phase, the preliminary result obtained from the threshold channel estimation algorithm is utilized as input to the trained neural network. By feeding this preliminary result into the network, the algorithm produces a more accurate output for channel estimation.

In [25], a transformer based on a self-attentive mechanism was first proposed. As shown in Figure 3, the Transformer estimation structure consists of two parts: the encoder and the decoder. To balance performance and computational complexity, our Transformer estimation model retains the encoder–decoder structure of the original model and changes the number of encoder and decoder layers from six layers to two layers. 

In the encoder, the time series data $\left[h_{t-L},h_{t-L+1},\cdots ,h_{t-1}\right]$ are mapped into multiple high-dimensional vectors through the feedforward fully connected layer at the input, where L denotes the time step. Subsequently, the temporal position information of the series is encoded in the position encoding area. The role of position encoding is to provide position information to the model through the linear variation of sin and cos functions. The positional encoding can be represented as
(12)$$PE(pos,2i)=\sin(pos/{10,000}^{2i/d_{\mathrm{model}}})$$
(13)$$PE(pos,2i+1)=\cos(pos/{10,000}^{2i/d_{\mathrm{model}}})$$
where pos represents the location index of information for each moment of the series, i represents the dimensional index of high-dimensional vectors obtained by mapping the time series through the input layer, and $d_{\mathrm{model}}$ is the dimension of high-dimensional vectors. 

Then, the positional encoding information is added to the output of the input layer. Concatenating these vectors together yields a matrix, $H^{\prime }=[h_{t-L+1}^{{}^{\prime }},h_{t-L}^{{}^{\prime }},\cdots ,h_{t}^{\prime }]$, as the input, and three vectors are generated from $H^{\prime }$ at each moment, which are the query vector, q, key vector, k, and value vector, v [26]:(14)$$\begin{array}{l} q_{t}=W^{q}h_{t}\\ k_{t}=W^{k}h_{t}\\ v_{t}=W^{v}h_{t} \end{array}$$
where W denotes the weighting matrix. The vectors at each moment are spliced into matrices: Q,K,V.

Matrices Q,K,V are calculated by the self-attentive mechanism to obtain the estimation results:(15)$$Attention(Q,K,V)=softmax(\frac{QK^{T}}{\sqrt{d_{k}}})V$$

The multi-headed attention mechanism is the integration of multiple independent attention modules, which is similar to multiple convolutional kernels in convolutional neural networks (CNN) to help the network extract richer features:(16)$$MultiHead(Q,K,V)=Concat(head_{1},\cdots ,head_{i})W_{o}$$
where Concat(•) denotes the matrix splicing operation, and $W_{o}$ denotes the weighting matrix. Finally, the residual network is applied to solve the problems of gradient disappearance and weight matrix degradation.

The results obtained from the multi-attention mechanism are subjected to residual connection layer and layer normalization operations to obtain the final outputs. In [27], ResNet was proposed to solve the problem of the difficult optimization of multilayer neural networks. The input–output relationship of the ResNet can be written as
(17)$$G\left(x\right)=F\left(x\right)+x$$
where x denotes the input, and $F\left(x\right)$ represents the nonlinear variation function. Compared to the non-residual network, when using the ResNet to calculate a partial derivative of x, a constant term is added to the derivation result, which keeps the model from losing gradient during training. Layer normalization serves to normalize all dimensions of each input sequence, thus speeding up model convergence and alleviating the gradient dispersion problem in deep network engineering.

### 3.2. Training Process

The proposed Transformer estimation algorithm training process can be described as follows: The inputs of the training data are the results of the conventional channel estimation algorithm, and the labels are actual channel responses. Because the channel response is a complex form, our training data splices the real and imaginary parts together. The channel responses from the previous L moments are used to predict the output of the next moment as follows:(18)$$h_{t}=f_{Transformer}(h_{t-L+1},\cdots ,h_{t-1})$$
where the timestep, L, can be determined by using the partial autocorrelation coefficient (PACF). The PACF is the linear correlation of the sequence, $\left\{h_{t}\right\}$, with the sequence $\left\{h_{t-k}\right\}$, with lag of order, k, and removes the linear dependence of $\left\{h_{t-1},h_{t-2},\cdots ,h_{t-(k-1)}\right\}$. In Figure 4, the magnitude of the correlation coefficients of the PACF is compared between the real parts and imaginary parts at different lags of order, k. The black solid line and the red dashed line in the Figure 4 represent the lags of order and the correlation coefficient when the data is approximately uncorrelated, respectively. It is clear to see that the PACF value drops below 0.1 when lags of order k=18, so we can assume that the correlation is almost nonexistent. It is worth mentioning that choosing a lag that is too short may result in incomplete learning of the temporal correlation within the sequence. Conversely, an excessively large lag could introduce extraneous noise during training, thereby impacting its effectiveness.

In the training phase, the mean square error (MSE) is selected as the loss function [28]:(19)$$J_{MSE}=\frac{1}{L}\sum_{i=1}^{N}(f_{Transformer}(h_{t-L+1},\cdots ,h_{t-1})-H_{label}{)}^{2}$$
where $H_{label}$ is the actual value and N is the total number of samples in the training set. The adaptive moment estimation (ADAM) algorithm is applied to update our dataset adaptively [29]. The sizes of our training set and test set are 8000 and 2000, respectively. The learning rate, batch size, and epochs are adjusted to 0.001, 128, and 150.

## 4. Performance Evaluation

### 4.1. NMSE and BER Performance

In this section, the simulation results of our proposed algorithm are given to verify the better performance than the threshold method and other deep learning methods. The parameters of our simulation are given in Table 1. 

The NMSE of our channel estimation method is computed as
(20)$$NMSE=\frac{{\left\|H^{\prime }-H\right\|}^{2}}{{\left\|H\right\|}^{2}}$$
where $H^{\prime }$ denotes the predicted value, and H is the true value.

In Figure 5, we compared the NMSE performance at different SNR when the $SNR_{p}$ is fixed at 30 dB, where $SNR_{p}=E_{pilot}/E_{data}$. The NMSE curves are close to a horizontal line because the transmitter power is constant, which leads to a reduction in the impact of the SNR. Furthermore, we can observe that the performance of DL methods is better than that of the threshold method. Among these neural networks, the performance of RNN and LSTM exceeds that of DNN. This is because RNN and LSTM have a powerful time series processing capability that DNN does not provide. LSTM slightly outperforms RNN due to the short-term dependency bottleneck inherent in RNN. The NMSE performance of our proposed Transformer estimation method exceeds the threshold method by approximately 18 dB. The Transformer estimation method outperforms LSTM due to its utilization of the multi-headed attention mechanism. This mechanism correlates the input sequences of the encoder with those of the decoder, enabling the model to focus on the encoder input sequences that exhibit a higher correlation with the predicted outputs. As a result, it reduces the impact of irrelevant information and enhances the prediction performance.

In Figure 6, the SNR is fixed at 25 dB, and the NMSE performance is compared at different $SNR_{p}$. It is evident that the channel estimation performance improves as the $SNR_{p}$ increases. This is because in the case of the lower $SNR_{p}$, the interference of noise is extremely serious, thus causing the threshold method to fail. We have observed that even at lower $SNR_{p}$, our proposed scheme surpasses the performance of the threshold scheme in terms of channel estimation. For example, the NMSE of the threshold scheme is about 12 dB when the $SNR_{p}$ = 30 dB, while our Transformer estimation model requires a $SNR_{p}$ of only 10 dB to achieve similar performance, which allows for a reduction in pilot power overhead.

Figure 7 illustrates the NMSE capability of the threshold scheme and other neural networks at different speeds. We can observe a decrease in NMSE performance with increasing velocity; this is because the channel we utilize changes at a slower rate in low-speed scenarios, and the channel estimation results are expected to be slightly superior compared to rapidly changing channels in high-speed scenarios. Although the NMSE performance decreases, the variety is not significant, only about 1~2 dB per 100 km/h; this is due to the good robustness of the OTFS system to high Doppler spread. 

In Figure 8, we show the BER comparison under different SNR, where the case of perfect channel estimation is also taken into account. It is observed that the BER decreases with increasing SNR. In the case of low SNR, the impact of noise can hinder the ability to accurately reflect the difference in channel estimation performance between individual algorithms. As SNR increases, the performance of the Transformer estimation algorithm acquires acceptable BER results. The threshold method exhibits the worst BER due to its inferior channel estimation performance. 

### 4.2. Computational Complexity

Space and time are two important metrics for evaluating the computational complexity of a DL model. In this paper, the quantity of parameters is chosen as the space evaluation metric, and the quantity of floating-point operations (FLOPs) is selected for the time aspect. As the number of parameters in a model increase, it necessitates a larger amount of data to effectively train the model. However, this can lead to a higher risk of overfitting, where the model becomes overly specialized to the training data. Additionally, excessive time complexity can significantly prolong the training and prediction processes.

The quantity of parameters refers to how many parameters the model contains and directly ascertains the size of the model. Usually, the parametric quantities of a model are analyzed to determine the memory usage of a computer. Assuming that the DL models contain multiple hidden layers, the formula for calculating the quantity of parameters for DNN, RNN, LSTM, and Transformer can be written respectively as
(21)$$Q_{DNN}=n_{h}^{1}\times n_{i}+n_{h}^{1}+\sum_{i=2}^{I}\left(n_{h}^{i-1}\times n_{h}^{i}+n_{h}^{i}\right)+n_{h}^{I}\times n_{o}+n_{o}$$
(22)$$Q_{RNN}=\left(n_{h}^{1}+n_{i}\right)\times n_{h}^{1}+n_{h}^{1}+\sum_{i=2}^{I}\left[\left(n_{h}^{i-1}+n_{h}^{i}\right)\times n_{h}^{i}+n_{h}^{i}\right]+n_{h}^{I}\times n_{o}+n_{o}$$
(23)$$Q_{LSTM}=\left[\left(n_{h}^{1}+n_{i}\right)\times n_{h}^{1}+n_{h}^{1}\right]\times 4+4\times \sum_{i=2}^{I}\left[\left(n_{h}^{i-1}+n_{h}^{i}\right)\times n_{h}^{i}+n_{h}^{i}\right]+n_{h}^{I}\times n_{o}+n_{o}$$
where I denotes the number of hidden layers, and $n_{i},n_{o},n_{h}^{1},n_{h}^{i}$ represent the number of neurons in the input layer, the output layer, the first hidden layer, and the *i*-th hidden layer, respectively.

The Transformer estimation model constructed in Section 3.1 is then analyzed for spatial complexity. First, a matrix transformation in a fully connected layer that does not contain activation operations can be written as
(24)$$R_{o}=T_{i}W+B$$
where $R_{o}\in^{S\times O}$, $T_{i}\in^{S\times I}$ denotes the output and the input of the network at that layer, respectively. W represents the weighting matrix, and B represents the bias matrix. Then, the number of parameters for a fully connected layer can be written as
(25)$$Q=\left(I+S\right)\times O$$

The quantity of FLOPs can be understood as the amount of computation, which is used to measure the complexity of the model time. Individual FLOPs generally refer to a single addition, subtraction, multiplication, or division operation. In neural networks, since the number of operations of the activation function is very small, it is not considered in FLOPs. Assuming that the DL models contain multiple hidden layers, the formula for calculating the quantity of FLOPs for DNN, RNN, LSTM, and Transformer can be written, respectively, as
(26)$$F_{DNN}=2n_{i}\times n_{h}^{1}+\sum_{i=2}^{I}\left(2n_{h}^{I-1}\times n_{h}^{I}\right)+2n_{h}^{I}\times n_{o}$$
(27)$$F_{RNN}=\left(\left(n_{i}+n_{h}^{1}\right)\times n_{h}^{1}\right)\times 2+2\times \sum_{i=2}^{I}\left[\left(n_{h}^{I-1}+n_{h}^{I}\right)\times n_{h}^{I}\right]+2n_{h}^{I}\times n_{o}$$
(28)$$F_{LSTM}=\left(\left(n_{i}+n_{h}^{1}\right)\times n_{h}^{1}\right)\times 2\times 4+4\times 2\times \sum_{i=2}^{I}\left[\left(n_{h}^{I-1}+n_{h}^{I}\right)\times n_{h}^{I}\right]+2n_{h}^{I}\times n_{o}$$

The number of FLOPs for a fully connected layer in the Transformer estimation model can be written as
(29)F=S×2IO
where the significance of each parameter is consistent with the space complexity analysis, so we do not explain it in detail.

Table 2 shows the complexity analysis of different neural network models. The 32-bit system and NVIDIA V100 with 14.13 TFLOPs, manufactured by the American company NVIDIA Corporation headquartered in Santa Clara, California, United States, are used as examples. As observed from the table, the number of parameters required by the Transformer estimation model is fewer than that of the LSTM and RNN. The higher parameter requirement in RNN and LSTM can be attributed to the inclusion of self-connecting feedback loops. These loops necessitate processing information from previous time steps before computing the hidden layer state for the next time step. Consequently, the historical hidden layer state information needs to be preserved until all time step inputs are processed, resulting in a substantial memory overhead. On the other hand, the Transformer estimation model eliminates the need for such sequential dependencies, leading to a smaller parameter count. The Transformer estimation model enables parallel processing of the input time series by using a multi-head attention mechanism, which greatly reduces the number of parameters in the model, and thus takes up only a small amount of memory space. A comparison of the FLOPs of the different models shows that the Transformer estimation model has the highest FLOPs. The higher FLOPs in the Transformer estimation model can be attributed to its extensive use of matrix transformations. These transformations involve a larger number of parameters and consequently result in longer computational time. However, this increased computational complexity allows the Transformer estimation model to effectively capture complex patterns and dependencies in the data, leading to superior performance in various tasks. Despite the additional computational requirements, the benefits of the Transformer estimation model make it a compelling choice.

## 5. Conclusions

In this paper, we focus on exploring the channel estimation algorithm for OTFS systems and propose a novel approach that incorporates the Transformer model. The inclusion of Transformer in our work stems from its exceptional capability to handle time series data more effectively when compared to other neural network architectures. Initially, we employ the threshold method to obtain preliminary channel estimation results. However, in order to achieve superior channel estimation performance, we consider the inherent temporal correlation present in channel responses. To leverage this correlation, we treat the data as time series and utilize them for predicting channel responses. The proposed Transformer estimation algorithm demonstrates better NMSE and BER performance than the threshold method proposed in the literature and conventional neural networks under the same SNR, $SNR_{p}$, and velocity. Moreover, through simulations, we validate the robustness of OTFS in scenarios with high Doppler spread. Additionally, our work highlights the potential of employing time series-based channel response prediction methods for accurate channel estimation. We also take into account the time complexity and spatial complexity of different models, where our proposed algorithm achieves the highest accuracy while maintaining acceptable computational complexity. Overall, our findings contribute to advancing the field of channel estimation in OTFS systems, demonstrating the effectiveness of incorporating Transformer models and highlighting their benefits in handling time series data.

## Figures and Tables

**Figure 1 entropy-25-01423-f001:**
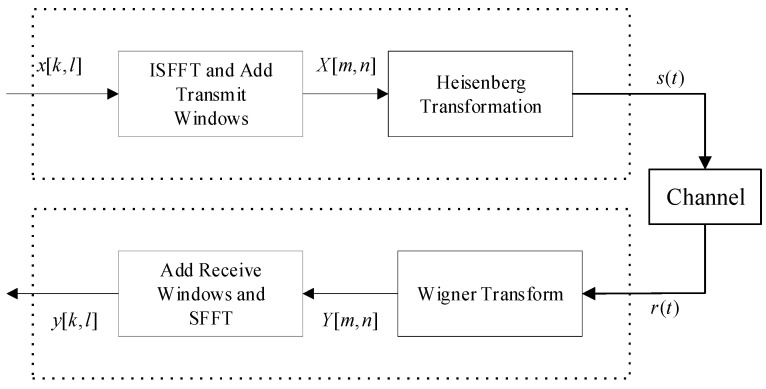
OTFS Modulation.

**Figure 2 entropy-25-01423-f002:**
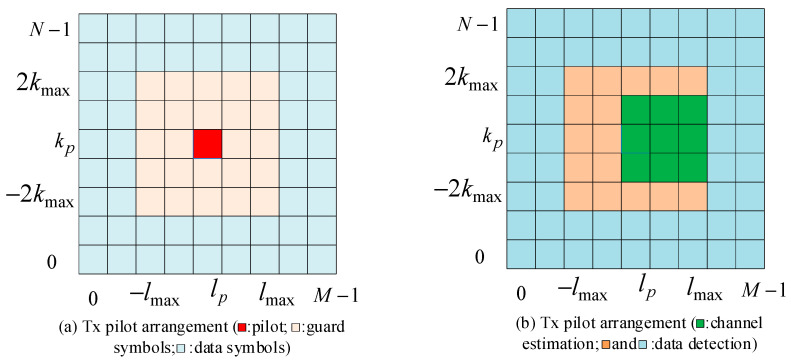
Pilot placement scheme.

**Figure 3 entropy-25-01423-f003:**
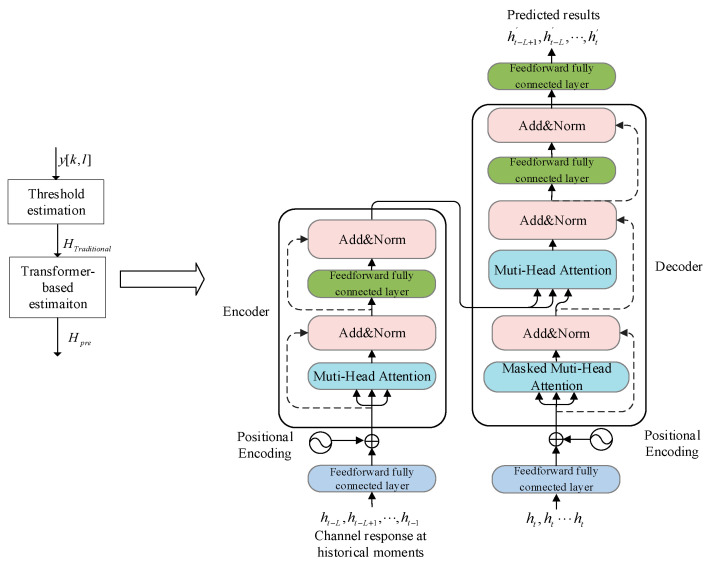
Transformer estimation structure.

**Figure 4 entropy-25-01423-f004:**
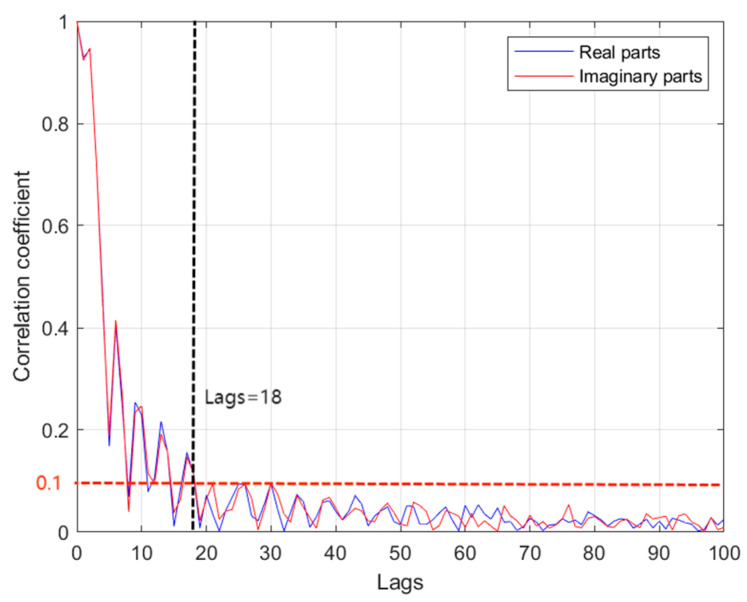
PCAF results for real parts and imaginary parts.

**Figure 5 entropy-25-01423-f005:**
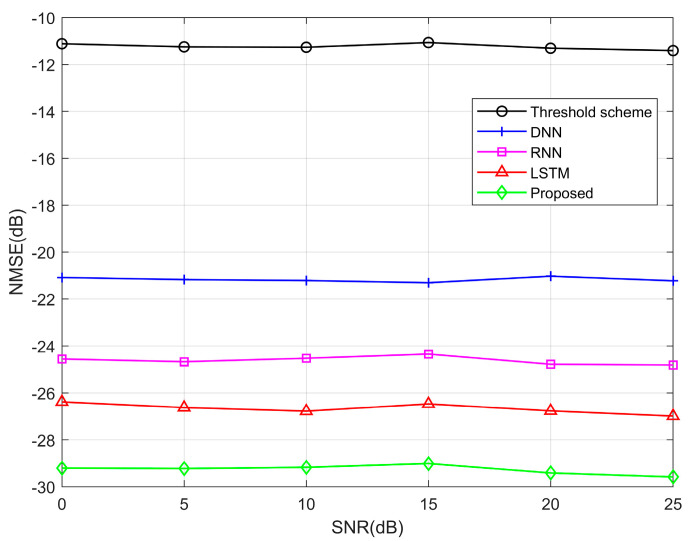
NMSE performance curves between different algorithms.

**Figure 6 entropy-25-01423-f006:**
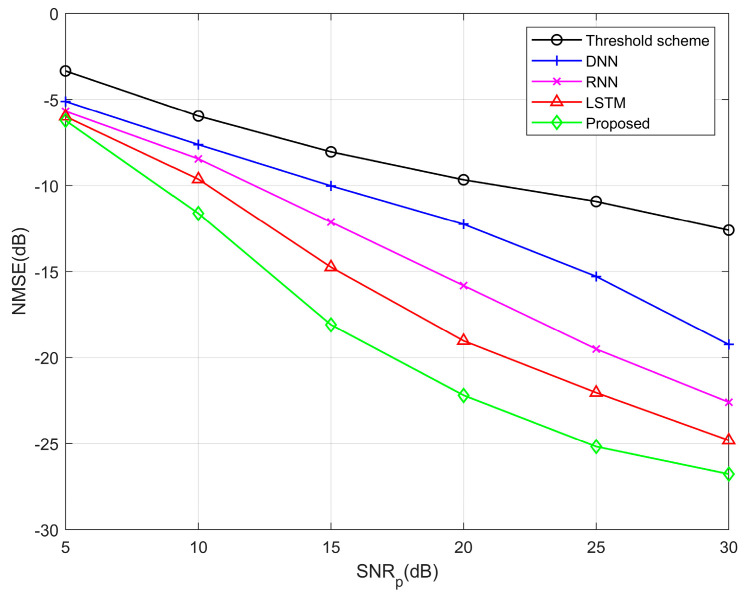
NMSE performance curves between different algorithms under different *SNR_p_*.

**Figure 7 entropy-25-01423-f007:**
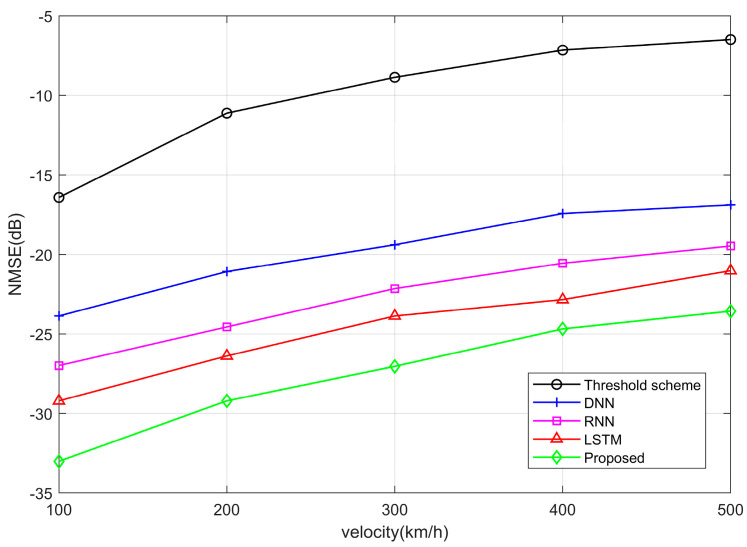
NMSE performance curves between different algorithms under different velocities.

**Figure 8 entropy-25-01423-f008:**
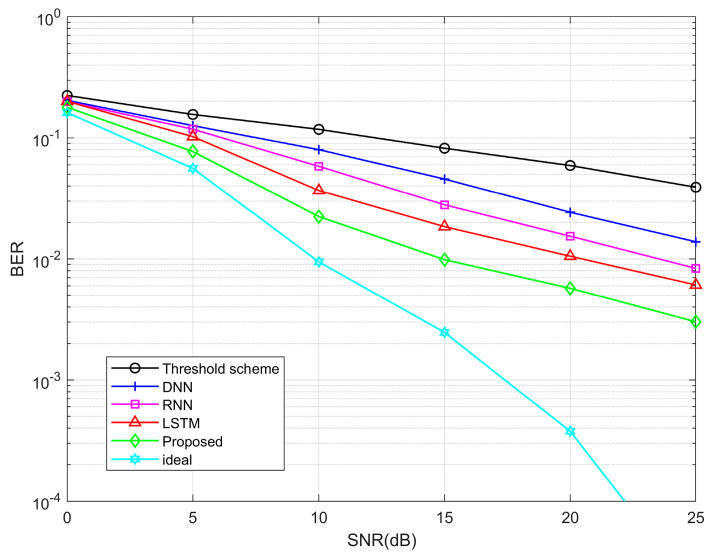
BER performance curves between different algorithms.

**Table 1 entropy-25-01423-t001:** Simulation parameters.

Parameter	Value
Carrier frequency	3.35 GHz
No. of subcarriers (M)	32
No. of OTFS symbols (N)	32
No. of channel paths	3
Subcarrier spacing (∆*f*)	15 kHz
Channel model	Rayleigh Channel
Path delay	[0, 0.2, 0.4] µs
Path power	[0, −10, −10] dB
Modulation alphabet	4-QAM

**Table 2 entropy-25-01423-t002:** Complexity analysis of different DL models.

DL Model	Space Complexity		Time Complexity	
	Number of Parameters	Memory Usage Size (MB)	Number of FLOPs	Computational Time (μs)
DNN	11,392	0.043	30,720	2.17 × 10^−9^
RNN	37,154	0.141	70,989	5.02 × 10^−9^
LSTM	148,609	0.567	354,596	2.51 × 10^−8^
Transformer estimation	14,077	0.0537	935,680	6.62 × 10^−8^

## Data Availability

Data sharing is not applicable.

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
