# Peer review of "A Transformer-Based Channel Estimation Method for OTFS Systems"

_entropy, 2023, doi:10.3390/e25101423_

Round 1

Reviewer 1 Report

In this manuscript, the authors have presented a Transformer-based channel estimation method for OTFS systems. Their results demonstrate that the proposed scheme outperforms the threshold method and other deep learning (DL) methods in terms of normalized mean squared error and bit error rate performance. Generally, the topic is timely, and the manuscript is clearly written and well organized. However, I have some major concerns as followings:

1. The preliminary channel estimation results are required in the proposed transformer-based scheme. The reviewer is wondering if the proposed transformer-based channel estimation scheme still works by using the original data instead of the preliminary channel estimation inputs. It is also suggested to highlight the novelty and advantages of the proposed scheme compared to other deep learning channel estimation methods.

2. The input-output relationship in (8) is impractical with ideal assumptions such as bi-orthogonal pulses and on-the-grid channel delays and Doppler shifts. Therefore, the merit of the proposed approach is not convincing, and the reviewer doubts the practical applicability of the method in real systems as shown in [R1] with non-ideal pulse-shapes and off-grid channel delays/Doppler shifts.

[R1] “Receiver design for OTFS with a fractionally spaced sampling approach,” IEEE Trans. Wireless Commun., vol. 20, no. 7, pp. 4072–4086, Jul. 2021.

3. The LMMSE detector is considered in this work with high complexity. Note that more advanced Memory AMP detector is proposed recently in [R2] with desired performance and low complexity. It is suggested to further discuss and elaborate on this.

[R2] “Low complexity memory AMP detector for high-mobility MIMO-OTFS SCMA systems,” in Proceedings of IEEE ICC Workshop OTFS-DDMC- 6G 2023.

4. In Fig. 7, this reviewer doesn’t understand why the performance degraded as the velocity increases. Intuitively, OTFS is resilience to different velocities (Doppler spreads).

5. In Fig. 8, there is a large BER gap between the perfect CSI scenario and the proposed scheme. Please explain and comment.

6. Generally speaking, joint channel estimation and data detection is more efficient as shown in [R3]. It is suggested to add some future plans for potential performance improvement by further considering joint channel estimation and data detection deep learning design for OTFS systems.

[R3] “Joint channel estimation and data detection for hybrid RIS aided millimeter wave OTFS systems,” IEEE Trans. Commun., vol. 70, no. 10, pp. 6832–6848, Oct. 2022.

7. The simulation results look solid for the chosen system parameters. However, it is not clear whether the performance is consistent across different system parameters such as different M and N, etc.

8. The reference format is not suitable and consistent for this journal. Please check and fix.

The manuscript is clearly written and well organized.

Reviewer 2 Report

The author presented good work in A Transformer Based Channel Estimation Method for OTFS Systems. The results are satisfactory. However, there are some concerns that should be addressed:

1)      The novelty of this paper over prior art is not clear. You must clarify it more with comparisons with previous research.

2)      The figures of the simulation results should be redrawn in a good way (resolution, legend, etc.).

3)      There are some grammatical issues in the text and some typos.

4)      Can the author compute EVM values from the BER ones in order to know the quality of the proposed transformer-estimation algorithm?

5)      Only the BER curve is shown. Can the authors comment on the EVM curve´s performance over OSNR?

6)      The OTFS system is achieved through simulation software; is it possible to validate it experimentally?

There are some grammatical issues in the text and some typos.

Minor editing of English language required

Round 2

Reviewer 1 Report

I have no further comments and suggest to accept the paper in its current version.

Reviewer 2 Report

Try to improve the quality of figures (font, size,...)

Try to enhance slightly the language